# Two-step mixed model approach to analyzing differential alternative RNA splicing

Li Luo[1,2☯], Huining Kang[1,2☯]*, Xichen Li[3], Scott A. Ness[1,2], Christine A. Stidley[1]

**1** Department of Internal Medicine, University of New Mexico, Albuquerque, New Mexico, **2** University of New Mexico Comprehensive Cancer Center, Albuquerque, New Mexico, **3** Department of Mathematics and Statistics, University of New Mexico, Albuquerque, New Mexico

☯ These authors contributed equally to this work.
* HuKang@salud.unm.edu

**Data Availability Statement:** All data and the R code is publicly available to the research community at the Dryad Digital Repository (DOI: 10.5061/dryad.66t1g1k0h). A copy of the data sets

## Abstract

Changes in gene expression can correlate with poor disease outcomes in two ways: through changes in relative transcript levels or through alternative RNA splicing leading to changes in relative abundance of individual transcript isoforms. The objective of this research is to develop new statistical methods in detecting and analyzing both differentially expressed and spliced isoforms, which appropriately account for the dependence between isoforms and multiple testing corrections for the multi-dimensional structure of at both the gene- and iso-form- level. We developed a linear mixed effects model-based approach for analyzing the complex alternative RNA splicing regulation patterns detected by whole-transcriptome RNA-sequencing technologies. This approach thoroughly characterizes and differentiates three types of genes related to alternative RNA splicing events with distinct differential expression/splicing patterns. We applied the concept of appropriately controlling for the gene-level overall false discovery rate (OFDR) in this multi-dimensional alternative RNA splicing analysis utilizing a two-step hierarchical hypothesis testing framework. In the initial screening test we identify genes that have differentially expressed or spliced isoforms; in the subsequent confirmatory testing stage we examine only the isoforms for genes that have passed the screening tests. Comparisons with other methods through application to a whole transcriptome RNA-Seq study of adenoid cystic carcinoma and extensive simulation studies have demonstrated the advantages and improved performances of our method. Our proposed method appropriately controls the gene-level OFDR, maintains statistical power, and is flexible to incorporate advanced experimental designs.

## Introduction

Gene expression profiles have proved extremely useful for evaluating and identifying subgroups among cancer patients with similar overt phenotypes. For example, in pediatric leukemia, gene expression signatures originally identified through microarray analyses have been converted into FDA-approved assays for classifying patients into distinct risk categories [1–4]. However, conventional gene expression profiles do not account for the differences observed in

and R scripts is also available at: http://www.unm.edu/~kanghn/software.htm

**Funding:** This work was supported by grants from the National Institutes of Health (https://www.nih.gov/) National Cancer Institute (R01CA170250 to S.A.N.), National Institute of Dental and Craniofacial Research (R01DE023222 to S.A.N.), and was partially supported by UNM Comprehensive Cancer Center Support Grant NCI P30CA118100, the Biostatistics Shared Resource, and Analytical and Translational Genomics Shared Resource. The computational resources used in this work were provided by the University of New Mexico Center for Advanced Research Computing. The funders had no role in study design, data collection and analysis, decision to publish, or preparation of the manuscript.

**Competing interests:** NO authors have competing interests

expressed isoforms when alternative RNA splicing is analyzed. Alternative RNA splicing can generate dozens of distinct transcripts from individual genes [5], which greatly increases the diversity of the whole-transcript expression patterns and the number of variables for statistical analysis. Transcript isoform expression profiles have been demonstrated to provide more informative cancer signatures than standard gene expression profiles [6,7]. The regulation of differential alternative RNA splicing has been implicated to correlate with poor disease outcome in two aspects: transcript level differential expression [8], and differentially spliced isoforms, e.g. changes in relative abundance of isoform expression for a gene [9]. Analyzing across the whole genome presents numerous obstacles, especially when the data should be analyzed in the context of more advanced experimental designs, such as time-course, factorial or multiple confounding designs.

High-throughput next-generation sequencing technologies such as RNA-sequencing (RNA-Seq) provide powerful tools for transcriptome analysis, reconstruction and quantification and offer an unprecedented opportunity to discover novel genes, transcripts and splice variants underlying complex diseases. While high-throughput RNA-Seq has become a standard for quantifying whole-transcriptome patterns of gene expression, the statistical methods for analyzing alternative RNA splicing and investigating differential expressed/spliced isoforms are still in their infancy. The early statistical methods for RNA-sequencing analysis focused on gene level expression analysis and utilized count-based modeling strategies, such as DESeq [10], edgeR [11], PoissonSeq [12], baySeq [13] and SAMseq [14]. These count-based strategies are not applicable to transcript level data, where the alignments of read counts to overlapping transcripts are ambiguous. Cufflinks/Cuffdiff2 [15–17] provides a tool for reconstruction of transcripts and quantification of isoform expression, but can only be used to analyze differential expression at the gene or isoform level between conditions. There are other packages (e.g., DEXSeq [18]) designed to analyze alternative RNA splicing by differential exon usage analysis, but the biological interpretation at the exon level is difficult. Attempts have been made to develop differential splicing or differential gene regulation analysis approaches, but many drawbacks and limitations still exist and improvement is needed. For example, rSeq-Diff [19] can only analyze one sample per condition and cannot handle biological replicates; thus it cannot be generalized to representing the differences between biological conditions. Diffsplice [20] provides a tool to analyze the differences in relative abundance of isoforms between conditions, but it incorporates the entire set of isoforms for each gene and is not able to trace the subset of isoforms that have true splicing effects. Ballgown [21] provides a bridge using a linear model approach to connect isoform quantification tools and downstream statistical tools for more flexible statistical analysis. It was reported that most isoform switches are independent of somatic mutations, which could uncover novel signatures and provide new molecular targets for therapy [22]. Sebestyen et al. developed an iso-kTSP algorithm to detect significant isoform switches, which measure the consistency in changes of relative expression of transcripts from the same gene [22]. The IsoformSwitchAnalyzeR algorithm was developed for identification and visualization of isoform switches with predicted functional consequences (e.g. gain/loss of protein domains) [23,24]. Despite the recent algorithm developments, the dependence between isoforms is ignored which limits statistical power. In summary, although several statistical packages are available for analyzing isoform expression, none do an adequate job of detecting and analyzing both differentially expressed and spliced isoforms, especially in the context of advanced experimental designs.

It is critical to understand and appropriately handle multiplicity in high dimensional alternative RNA splicing analysis. Most current statistical methods simply apply multiple testing procedures such as the Benjamini and Hochberg procedure (BH) to control for the false discovery rate (FDR) at the levels of exons, isoforms, or genes by treating them as independent

features. The multiple comparison correction for RNA splicing analysis remains a challenge when evaluating all expressed isoforms, particularly for detection of isoform switches which evaluates all possible pairs [22,24]. The overall false discovery rate (OFDR) was introduced as an appropriate error rate to control [25], which is recommended over FDR because it focuses on the inferential units of interests (genes) and is more powerful by testing a much smaller number of screening hypotheses [26]. It was illustrated that controlling the FDR does not guarantee control of the OFDR, and similarly controlling for OFDR does not guarantee appropriate FDR control. A stage-wise method (stageR) was proposed to control the gene level false discovery rate (FDR) and boost power in analysis of differential transcript usage [27]. We applied the similar concept of controlling for OFDR in the multi-dimensional alternative RNA splicing analysis, where the OFDR is defined as the expected proportion of falsely discovered genes out of all discovered genes.

In this research, we developed a unified linear mixed effects model-based statistical approach accompanied by a two-step hierarchical hypothesis testing framework for analyzing complex alternative RNA splicing regulation patterns detected by whole-transcriptome RNA-sequencing technologies. We present a novel application of two-step hierarchical hypothesis testing procedure coupled with a linear mixed model setting to analyze the differential expression at the multi-dimensional gene- and isoform- level using data from RNA sequencing technologies. Our approach provides three key advantages. First, the linear model approach thoroughly characterizes and differentiates three types of genes related to alternative RNA splicing events as demonstrated in Fig 1, e.g. genes with (i) no differentially expressed isoforms; (ii) differential expression of isoforms but no differential splicing; and (iii) differentially spliced isoforms with differential expression at the isoform level but not necessarily at the gene level. Specifically, Fig 1 describes the pattern of log-scale isoform level expression for these three types of genes, which correspond to Models 0–2 proposed in Shi *et al.* [19]. The genes of type (iii) are of special interest, which implicate the presence of isoform switches. Second, our mixed model naturally incorporates the important features of biological correlation patterns between isoforms of the same gene, which have typically been ignored in other existing approaches. The third major advantage is that the employment of a two-step hierarchical hypothesis testing approach appropriately control the overall false discovery rate (OFDR) and yield greatly improved statistical power and computational efficiency by dramatically reducing the number of tests performed. To our knowledge, there is no existing statistical package that has incorporated all the aforementioned analyses at the same time.

We applied this approach to two whole transcriptome RNA-Seq studies of adenoid cystic carcinoma [28] and pediatric acute myeloid leukemia [29], and compared the results with other methods (Cuffdiff, Limma, and t-test with BH correction) in identification of differentially expressed/spliced isoforms. We conducted extensive simulation studies to evaluate the performance of our methods in controlling gene-level OFDR and achieving the power to detect true differences in alternative RNA splicing.

## Materials and methods

### A linear mixed effects model approach

We start with the raw sequence reads from the RNA-Seq experiment that have been preprocessed and for which we have obtained the expression estimates for the isoform (transcript) abundance measured in fragments per kilobase of transcript per million fragments mapped (FPKM) or transcripts per million (TPM). This can be done by using, for example, Cufflinks [15–17], kallisto [30] and RSEM [31]. Our interest is to identify isoforms that are differentially expressed or spliced among different conditions. To reach this goal we propose a linear mixed

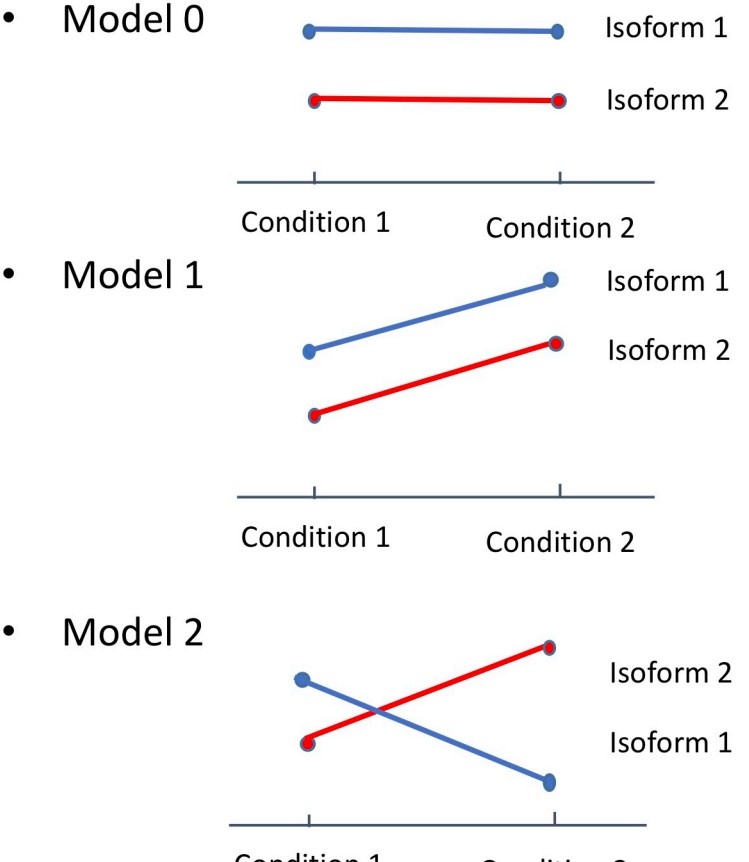

**Fig 1. Expression patterns of three types of genes with isoform abundance levels in log scale, which correspond to Models 0–2 proposed in Shi et al. [19].** (Model 0) no differentially expressed isoforms; (Model 1) differential expression of isoforms but no differential splicing; and (Model 2) differentially spliced isoforms with differential expression at the isoform level but not necessarily at the gene level.

effects model for the isoform abundance. Suppose we have $M$ genes and $J$ conditions. For the $j$th condition there are $K_j$ samples and the total number of samples is $N = K_1 + K_2 + \cdots + K_J$. For a given gene $m$ we assume that there are $L_m$ isoforms and we denote $Y_{jklm}$ as the abundance in the log scale of the $l$th isoform of gene $m$ in sample $k$ nested in condition $j$ ($j = 1,2,\ldots,J$, $k = 1,2,\ldots,K_j$, $l = 1,2,\ldots,L_m$, $m = 1,2,\ldots,M$). For simplicity in notation throughout the following, we assume we are discussing gene $m$ and drop the fourth subscript $m$. We assume $Y_{jkl}$ follows a normal distribution and can be modeled using the following linear mixed effects model for two-factor split-plot design with unequal variances among the effects of isoforms:

$$Y_{jkl} = \beta^G + \beta_l^I + \beta_j^C + \beta_{jl}^{IC} + \rho_{k(j)} + \varepsilon_{jkl}, \tag{1}$$

where, $\beta^G$ represents the baseline isoform expression in the log scale of the gene. $\beta_l^I$ is the logarithm of the expected relative expression of the $l$th isoform. $\beta_j^C$ is the logarithm of the fold change in overall expression of the given gene under condition $j$. $\beta_{jl}^{IC}$ is the effect that condition $j$ has on the relative expression of the $l$th isoform. $\rho_{k(j)}$ is the effect of the $k$th sample (whole

plot), nested within the $j$th condition and we assume that $\rho_{k(j)} \sim N(0, \ \sigma_\rho^2)$. $\varepsilon_{jkl}$ is random error and $\varepsilon_{kjl} \sim N(0, \ \sigma_I^2)$. $\rho_{k(j)}$ and $\varepsilon_{jkl}$ are mutually independent. The mixed model (1) contains the effects of conditions and isoforms and the condition by isoform interaction as fixed effects and the subject effects as random effects.

Let's denote the observations of the $k$th sample (whole plot), nested within the $j$th condition as $Y_{jk} = (Y_{jk1}, \ Y_{jk2}, \ \cdots, Y_{jkL})^T$. Then the covariance structure for $Y_{jk}$ is

$$Var(Y_{jk}) = \boldsymbol{\Sigma} = \begin{bmatrix} \sigma_1^2 + \sigma_\rho^2 & \sigma_\rho^2 & \cdots & \sigma_\rho^2 \\ \sigma_\rho^2 & \sigma_2^2 + \sigma_\rho^2 & \cdots & \sigma_\rho^2 \\ \vdots & \vdots & \ddots & \vdots \\ \sigma_\rho^2 & \sigma_\rho^2 & \cdots & \sigma_L^2 + \sigma_\rho^2 \end{bmatrix} \tag{2}$$

We denote (2) as the covariance structure with unequal error variances. The above covariance structure is reduced to compound symmetry when $\sigma_1 = \sigma_2 = \cdots = \sigma_L$, and is generalized to unstructured when there are no constraints on the covariance elements in (2).

**Null hypotheses.** To identify isoforms that are differentially expressed or spliced among different conditions, we will consider two types of hypothesis tests as described below:

Type 1: Identify genes whose isoforms are differentially expressed or differentially spliced (either Model 1 or Model 2 genes in Fig 1)

$$H_0: \ \beta_j^C = \beta_{jl}^{IC} = 0 \tag{3}$$

Type 2: Identify genes whose isoforms are differentially spliced only (Model 2 in Fig 1)

$$H_0: \ \beta_{jl}^{IC} = 0 \tag{4}$$

We first fit the linear mixed effects Model (1), and then perform likelihood ratio tests (LRT) of the coefficients to identify genes with differentially expressed/spliced isoforms.

**A two-step hierarchical hypothesis-testing framework.** Many existing approaches for detecting differentially expressed isoforms treat each isoform as a single genomic feature and perform the hypothesis tests for all isoforms simultaneously. These approaches usually lack statistical power due to the large number of hypothesis tests performed at the same time. A two-stage procedure was proposed to test differentially expressed gene sets [26], which was later generalized to a two-step hierarchical hypothesis set testing framework in microarray time-course experiments [32]. A similar two-step approach (stageR) was proposed to control the gene level false discovery rate (FDR) and boost power in analysis of differential transcript usage [27]. We will utilize a similar two-step hierarchical hypothesis-testing framework for identifying isoforms of interest. For genes, the first step of the proposed framework is to perform gene level tests to identify genes that have differentially expressed or spliced isoforms, which we refer to as the screening tests. In the second step we examine only the isoforms of genes that have passed the screening tests.

In the initial screening, we will test for two types of null hypotheses, i.e. (3) and (4), for identification of differentially expressed isoforms or only differentially spliced isoforms. In the subsequent step we test genes that passed the screening test to see whether each individual isoform is differentially expressed. We propose two options: the first option is to perform the test using either the t-test or one-way ANOVA; and the second option is to perform the test using the contrast (Wald-test) based on the linear mixed effects model. The null hypotheses for testing the differential expression of the $l$th isoform among $J$ conditions are described in the

following equation:

$$H_0 : \begin{cases} \beta_2^C - \beta_1^C + \beta_{2l}^{IC} - \beta_{1l}^{IC} = 0 \\ \beta_3^C - \beta_1^C + \beta_{3l}^{IC} - \beta_{1l}^{IC} = 0 \\ \vdots \\ \beta_J^C - \beta_1^C + \beta_{Jl}^{IC} - \beta_{1l}^{IC} = 0 \end{cases} \qquad (5)$$

Utilizing a two-step hierarchical hypothesis-testing framework, we will control for the overall false discovery rate (OFDR) at the gene level, which was recommended over FDR because it focuses on the inferential units of interest [26]. Compared with standard approaches that perform the hypothesis tests on all isoforms simultaneously, the number of tests performed in the proposed framework will be dramatically reduced, and consequently the statistical power is expected to be increased [32].

**Procedure to control for OFDR.** We control the OFDR which is defined as the expected proportion of

$$\text{falsely discovered genes out of all discovered genes: } = E\left(\frac{V}{R}\right), \qquad (6)$$

where R is the total number of rejected hypotheses or discovered genes, and V is the number of discovered genes where at least one null hypothesis (including screening and confirmatory hypotheses) was incorrectly rejected.

We aim to perform two levels of inferences while controlling for the OFDR, i.e. test for differentially expressed or differentially spliced genes and at the same time test for individual isoform differential expression. The specific procedure is described as below in details:

1. Screening stage: apply the Benjamini-Hochberg procedure [33] at level $\alpha$ to the p-values of the $M$ screening tests. Let $R$ be the number of genes that pass the screening tests.

2. Confirmatory stage: for each gene that passes the screening test, test the hypotheses for the $L$ individual isoforms simultaneously, applying a p-value adjusting procedure on the p-values of the $L$ tests such that the family-wise error rate (FWER) of these $L$ tests is controlled at level $R\alpha/M$. Specifically we will control the FWER using three methods: Bonferroni, Holm, and Hochberg methods.

It was proved that the above procedure controls the OFDR at level $\alpha$ under the condition that the individual hypothesis tests in the second confirmatory stage are independent from all other screening tests [26,32].

# Results

## Application to an RNA-sequencing study of adenoid cystic carcinoma (ACC)

We applied the proposed linear mixed model approach to a study of adenoid cystic carcinoma where RNA-sequencing was performed on 20 ACC salivary gland tumors and five normal salivary glands. Details of the RNA sequencing were described previously [28]. We present primary results using the covariance structure with unequal error variances as shown in Eq (2). Cufflinks was used to estimate and quantify the isoform abundance in FPKM prior to the differential expression/splicing analysis. We examined 2850 genes having two or more detected isoforms in our two-step analyses, and compared results using our proposed methods with

other widely used methods for analysis of RNA-Seq data: Cuffdiff [17], Limma, and t-test with BH correction.

We performed differential isoform expression/splicing analysis between 8 patients who are free of cancer vs. 6 patients who are not among whom have complete follow-up. The multi-dimensional multiple comparisons were corrected using our proposed procedure, and statistical significance was considered at OFDR = 0.10 due to the exploratory nature of this study where the sample sizes and effect sizes are limited. While no isoforms are called significantly different between the two patient outcome groups using the three alternative methods (Cuffdiff [17], Limma, and simple t-tests), our approach using the covariance structure with unequal error variances identified 11 genes that have either differentially spliced or differentially expressed isoforms using a simultaneous test for the main group effect along with the interaction effect (Table 1). In addition, a Type 2 screening test identified 4 genes that are differentially spliced using a screening test for the condition by isoform interaction term. A second stage confirmatory test using the Hochberg method further identified 12 isoforms from 9 genes that are differentially expressed (Table 2). The confirmatory test using the Bonferroni or Holm methods yielded the same results as the Hochberg method. We also performed the differential alternative RNA splicing analysis using the proposed linear mixed model with an unstructured covariance structure for the exploratory purpose. The limitation for specifying the unstructured covariance matrix lies in the possible inflated type I error rates due to the large number of covariance parameters included in the estimation process. For this reason, we performed the analysis at significance level FDR = 0.05 and we summarize the results in S1 and S2 Tables. As the result, 50 genes have passed the Type 1 screening test and 8 genes have passed the Type 2 screening test. Seven of the 11 genes identified in the type 1 screening test using the unequal variance covariance structure were also identified by assuming unstructured covariance variance, indicating agreement between the two choices of covariance matrix. We observed that the p-values of the first step gene-based screening test using both covariance matrix structures are highly correlated with each other (Spearman's $\rho$ of 0.92 and 0.94 for type I and type II screening tests).

For each gene, the group mean values of the log-transformed expression of different isoforms in FPKM detected and estimated by Cufflinks are plotted in Fig 2. Groups (free of

**Table 1. List of genes that are called significant between 8 patients who are free of cancer vs. 6 patients in the screening tests along with the likelihood ratio test p-values and FDR.** Type 1 screening test identified 11 differentially expressed/spliced genes, and Type 2 screening test identified 4 differentially spliced genes. False discovery rate was controlled at the 0.10 level.

| Gene name | Type 1 screening | | Type 2 screening | |
|---|---|---|---|---|
| | P value | FDR | P value | FDR |
| POSTN | 9.14E-06 | **0.03** | 0.00233 | 0.36 |
| HNRNPA2B1 | 0.00004 | **0.06** | 0.00022 | 0.12 |
| VEGFA | 0.00012 | **0.09** | 0.43157 | 0.84 |
| FRMD4A | 0.00017 | **0.09** | 0.06945 | 0.54 |
| TRIP12* | 0.00020 | **0.09** | 0.00012 | **0.09** |
| PRKAA1* | 0.00021 | **0.09** | 0.00012 | **0.09** |
| GNPTAB | 0.00023 | **0.09** | 0.01255 | 0.44 |
| C3orf17* | 0.00030 | **0.09** | 0.00012 | **0.09** |
| ASXL1 | 0.00030 | **0.09** | 0.00425 | 0.43 |
| RRBP1* | 0.00034 | **0.09** | 0.00010 | **0.09** |
| FZD6 | 0.00037 | **0.09** | 0.00104 | 0.24 |

* Genes that passed both Types 1 and 2 screening tests.

**Table 2. List of 12 differentially expressed isoforms between 8 patients who are free of cancer vs. 6 patients in the confirmatory test.**

| Isoform ID | gene name | Fold Change | P value (model-based) | P value (t test) | Two step significance threshold |
|---|---|---|---|---|---|
| ENST00000474311 | C3orf17 | 5.50 | 8.50E-07 | 0.00030 | 3.40E-06 |
| ENST00000492155 | FRMD4A | 3.67 | 7.35E-06 | 0.00178 | 2.94E-05 |
| ENST00000358621 | FRMD4A | 2.01 | 9.13E-06 | 0.00137 | 3.65E-05 |
| ENST00000522566 | FZD6 | 8.59 | 8.12E-08 | 0.00028 | 1.62E-07 |
| ENST00000356674 | HNRNPA2B1 | 1.68 | 5.72E-11 | 0.00081 | 3.43E-10 |
| ENST00000541179 | POSTN | 19.31 | 8.16E-08 | 0.00087 | 4.08E-07 |
| NST00000478947 | POSTN | 9.48 | 4.80E-07 | 0.00037 | 2.40E-06 |
| ENST00000379743 | POSTN | 9.15 | 6.16E-05 | 0.00394 | 0.00031 |
| ENST00000511248 | PRKAA1 | 7.34 | 3.03E-08 | 0.00026 | 9.10E-08 |
| ENST00000495501 | RRBP1 | 5.89 | 2.49E-05 | 0.00139 | 7.46E-05 |
| ENST00000428959 | TRIP12 | -14.65 | 4.19E-08 | 0.00249 | 2.93E-07 |
| ENST00000497139 | VEGFA | 7.58 | 7.29E-07 | 0.00052 | 1.46E-06 |

cancer or not) are labeled on the X axis. The last six digits of the Ensemble transcript IDs are given in the legend. Under the null hypothesis that there is no differential splicing, the distributions of the relative isoform abundance are expected to be similar, and the lines representing

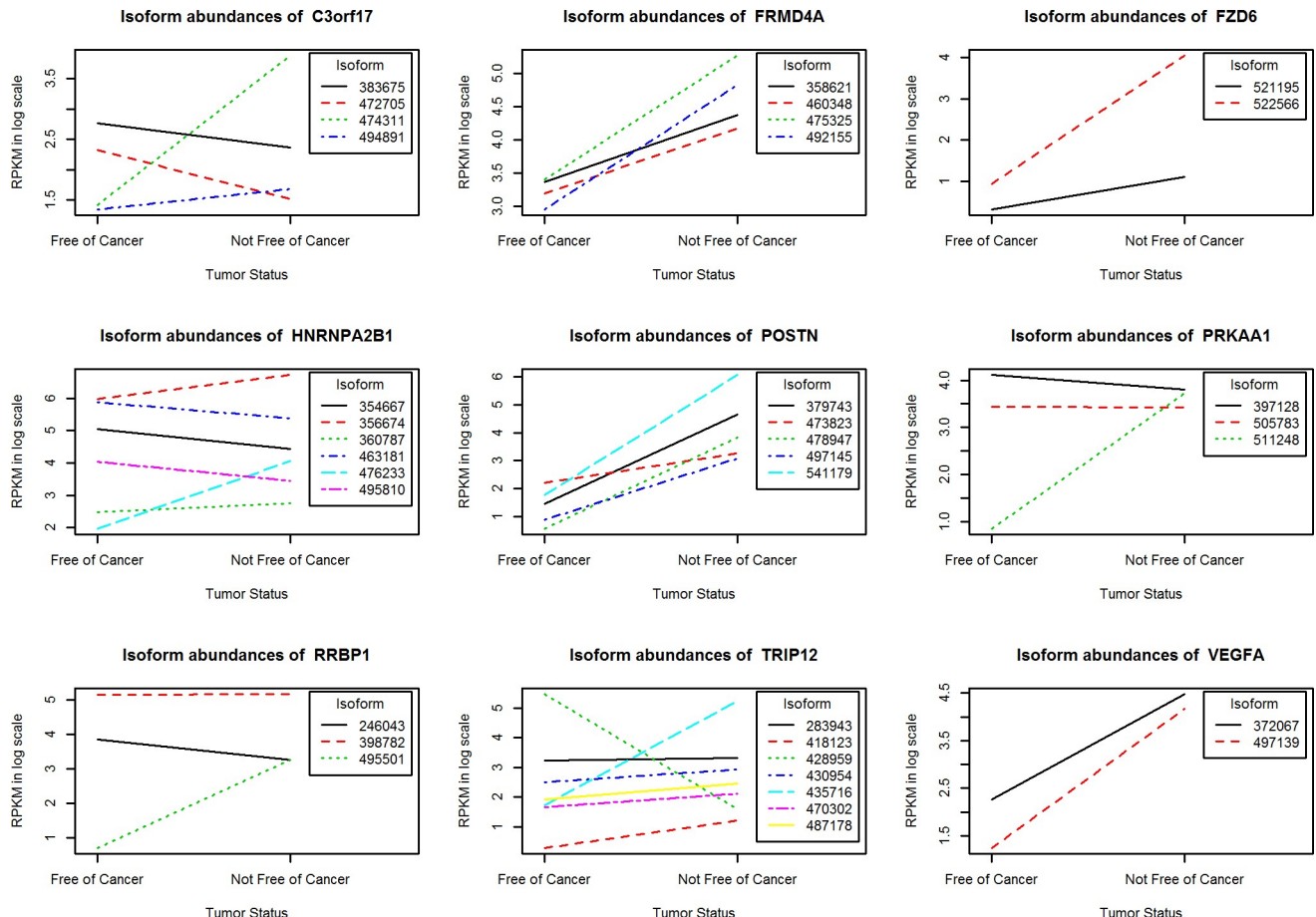

**Fig 2. The isoform expression profiles for nine differentially expressed/spliced genes between 8 patients who are free of cancer vs. 6 patients who are not.**

the mean isoform expression profile for the two conditions, normal vs. tumor, are expected to be parallel. We have observed considerable non-parallel patterns for the 4 differentially spliced genes that passed Type 2 screening test: *C3orf17*, *PRKAA1*, *RRBP1*, and *TRIP12*.

Our preliminary results have revealed genes with combined patterns of differential expressed/spliced isoforms. We observed the overexpression of isoforms of genes *C3orf17*, *FRMD4A*, *FZD6*, *HNRNPA2B1*, *POSTN*, *PRKAA1*, *RRBP1* and *VEGFA*, and the under expression of *TRIP12* isoform (ENST00000428959) in ACC patients who are not free of cancer, which is a surrogate for poor ACC outcomes (Table 2 and Fig 2). Up- or down- regulation of isoforms exist even for genes (e.g. *HNRNPA2B1*, *TRIP12*) which do not appear to be differentially expressed at the gene level. A number of these genes were shown in previous reports to be associated with cancer prognoses. The high expression level of the *POSTN* gene was repeatedly reported to correlate with poor outcome of different human malignancies, which include shorter progression-free survival following first-line chemotherapy in epithelial ovarian cancer [34], more advanced stage and lower survival rates in colorectal cancer patients [35], and high grade and invasive meningioma [36].

The overexpression of *VEGFA* was implicated in the poor outcome of breast cancer [37,38]. There is significant evidence that the *VEGFA* gene alternative RNA splicing plays an important role in tumor growth and progression, suggesting a potential target for new cancer therapies [39,40]. Colorectal cancer patients with high *RRBP1* expression had shorter disease specific survival compared to those with low *RRBP1* expression [41]. The up-regulation of *FRMD4A* was reported in human head and neck squamous cell carcinoma (HNSCC) and correlated with increased risks of relapse [42]. The over expression of *FZD6* was reported to be associated with poor prognosis in glioblastoma patients [43]. A recent RNA-Seq study of acute myeloid leukemia patients at remission identified an aberrant RNA splicing of exon3-skipping event in *TRIP12* suggesting future investigation of its use as a potential target [44].

## Application to an RNA-sequencing study of pediatric acute myeloid leukemia (AML)

We also applied the proposed approach to identify differentially expressed/spliced isoforms that are associated survival outcome in a large cohort of pediatric AML from the NCI/COG TARGET-AML initiative [29]. The RNA-seq data at exon, isoform and gene levels as well as the clinical data for the AML patients are downloaded from NCI/COG TARGET website at https://ocg.cancer.gov/programs/target. We analyzed a subset of 234 patients who have clinical outcomes with 81 cases (relapsed within 3 years) and 153 controls (CCR for at least 3 years). After removing those with low abundances and those that are not in the coding region, we analyzed a total of 35397 isoforms representing 8058 genes. The multi-dimensional multiple comparisons, again, were corrected using the two-step procedure, and statistical significance was considered at standard cutoff of. 0.05 for OFDR. With our approach using the covariance structure with unequal error variances, we identified 782 genes that have either differentially spliced or differentially expressed isoforms using the Type 1 screening tests that simultaneously test for the main group effect along with the interaction effect; and 857 genes that are differentially spliced using the Type 2 screening which tests for just the condition by isoform inter-action term. Table 3 summarizes the distribution of the genes detected using our method that passed either Type 1 or Type 2 screening tests grouped by the number of studied isoforms for each gene. The union of the two lists of genes is presented in S3 Table. A second stage confirmatory test using the t-test with Hochberg method for controlling FWER further identified 269 and 203 isoforms (S4 and S5 Tables) that are differentially expressed based on the aforementioned two lists of the genes, respectively. Note that the majority of the genes from Type 2 screening test cannot be

**Table 3. The distribution of the number of significant genes that passed Type 1 or Type 2 screening tests comparing the isoform expression patterns between 81 cases and 153 controls.**

| No. of isoforms | 2 | 3 | 4 | 5 | 6 | 7 | 8 | 9 | 10 | 11 | 12 | 13 | 14 | 15 | ≥16 | Total |
|---|---|---|---|---|---|---|---|---|---|---|---|---|---|---|---|---|
| No. of genes that passed Type 1 screening test | 88 | 111 | 105 | 90 | 88 | 67 | 69 | 46 | 27 | 31 | 20 | 14 | 10 | 4 | 12 | 782 |
| No. of genes that passed Type 2 screening test | 100 | 123 | 116 | 98 | 101 | 68 | 75 | 52 | 29 | 34 | 21 | 14 | 10 | 5 | 11 | 857 |

identified by the traditional gene level differential expression analysis. We performed the t-test with BH adjustment on the gene expression data of these 857 genes and none of them are called significant at level FDR = 0.05. Fig 3 shows the isoform expression profiles of two genes, EEF1B2 and RUNX2. While there seems no difference in the overall expressions of the genes there appears to be interesting splicing events for these two genes. EEF1B2 is one of the Eukaryotic translation factors that have received much attention recently with regards to their role in the onset and progression of different cancers [45]. For this gene, two isoforms ENST00000435123 and NEST00000455150 are significantly up- and down-regulated in cases as compared to controls while the rest of 6 isoforms do not seem to differ between the two conditions. Another gene RUNX2 is a member of RUNX family proteins that are generally considered to function as a tumor suppressor in the development of leukemia. Our approach has identified one isoform NEST00000371436 that is significantly down-regulated in the cases as compared to the control while the rest of the isoforms showed no difference between the two conditions. As a comparison to our approach we applied t-tests with BH multiple comparison adjustment to the same data set. We found that only 13 isoforms (representing 12 unique genes) were significantly differentially expressed between the two conditions at the level FDR = 0.05. We also performed the differential alternative RNA splicing analysis using the proposed linear mixed model with an unstructured covariance structure and the results can be found at http://www.unm.edu/~kanghn/software/. We observed that the p-values of the first

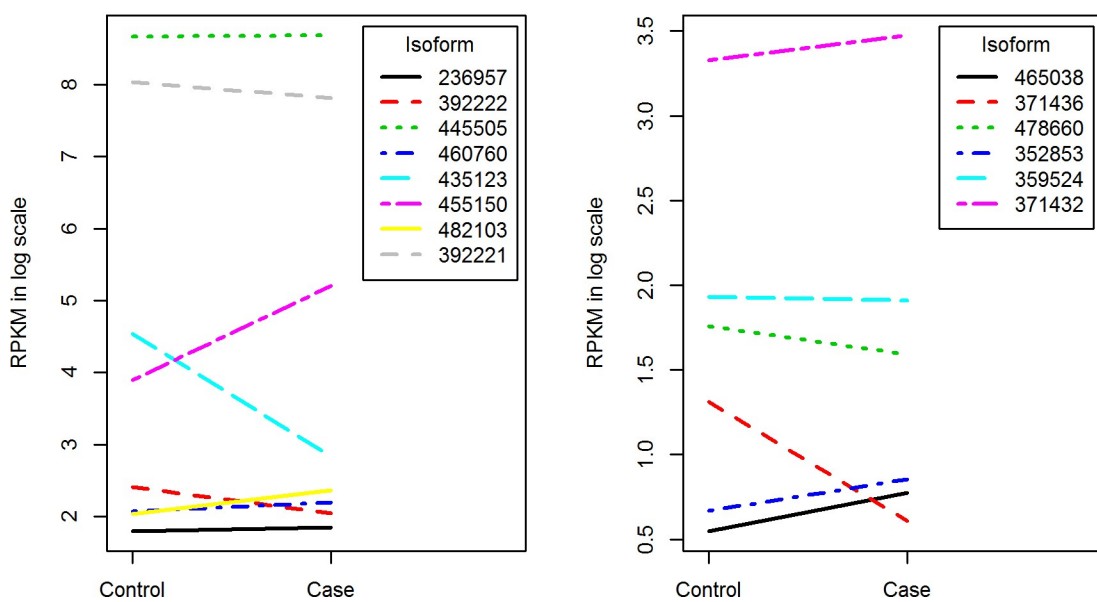

**Fig 3. Isoform expression profiles for two gene with differentially spliced isoforms that are associated with survival outcome in AML.**

step gene-based screening test using both covariance matrix structures are highly correlated with each other (Spearman's $\rho$ of 0.92 and 0.94 for type I and type II screening tests).

## A simulation study

We performed a simulation study to verify that our proposed two-step procedures with different options for the confirmatory stage inference properly control the OFDR and to compare their statistical powers. In order to capture the dependence structure of all the isoforms for each gene, we conducted the following simulations based on the data set used in Section 3.2, where we considered two conditions, free of cancer versus not. We considered Type 1 screening test and used the covariance structure with unequal error variances as shown in Eq (2). The results for Type 2 screening should be similar.

In the case of two conditions ($J = 2$) we re-write model (1) as the following (7) so that the model specification corresponds closely to the syntax used in R package nlme.

$$Y_{jkl} = \beta^G + \beta_2^C C_j + \beta_2^I I_{2l} + \cdots + \beta_L^I I_{Ll} + \beta_{22}^{IC} C_j I_{2l} + \cdots + \beta_{2L}^{IC} C_j I_{Ll} + \rho_{k(j)} + \varepsilon_{jkl}, \tag{7}$$

where we assume

$$\beta_1^C = \beta_1^I = \beta_{11}^{IC} = \cdots = \beta_{1L}^{IC} = \beta_{21}^{IC} = 0, \tag{8}$$

and $C_j$, $I_{l,l}$ are dummy variables such that $C_j = \begin{cases} 1, & \text{if } j = 2 \\ 0, & \text{if } j = 1 \end{cases}$ and $I_{l'l} = \begin{cases} 1, & \text{if } l = l' \\ 0, & \text{if } l \neq l' \end{cases}$ ($j = 1,2,$ $l = 1,\cdots,L, k = 1,\cdots,K_j$). The mean difference in abundance (in log scale) of isoform $l$ between the two conditions is

$$\mu_{2l} - \mu_{1l} = \begin{cases} \beta_2^C & \text{if } l = 1, \\ \beta_2^C + \beta_{2l}^{IC} & \text{if } l = 2, \cdots, L. \end{cases} \tag{9}$$

The matrix form of (7) can be written as

$$\boldsymbol{Y}_{jk} = X_{jk}\boldsymbol{\beta} + \boldsymbol{e}_{jk}, \tag{10}$$

where $\boldsymbol{Y}_{jk} = (Y_{jk1},\ Y_{jk2},\ \cdots, Y_{jkL})^T$, $\boldsymbol{\beta} = (\beta^G\ \beta_2^C\ \beta_2^I \cdots \beta_L^I\ \beta_{22}^{IC} \cdots \beta_{2L}^{IC})^T$,

$$X_{jk} = \begin{bmatrix} 1 & C_j & 0 & 0 & 0 & 0 & 0 & 0 & \cdots & 0 \\ 1 & C_j & 1 & 0 & 0 & 0 & C_j & 0 & \cdots & 0 \\ 1 & C_j & 0 & 1 & 0 & 0 & 0 & C_j & \cdots & 0 \\ \vdots & \vdots & \vdots & \vdots & \ddots & \vdots & \vdots & \vdots & \ddots & \vdots \\ 1 & C_j & 0 & 0 & \cdots & 1 & 0 & 0 & \cdots & C_j \end{bmatrix}, \tag{11}$$

and $\boldsymbol{e}_{jk} = (e_{jk1},\ e_{jk2},\ \cdots, e_{jkL})^T$ is a multivariate normal distribution random vector with a zero vector mean and a covariance matrix (2).

There are $2L$ parameters $\beta^G, \beta_2^C, \beta_2^I, \cdots, \beta_L^I, \beta_{22}^{IC}, \cdots, \beta_{2L}^{IC}$ corresponding to the fixed effects and $L + 1$ parameters in the variance-covariance matrix (2) for a full model, which we refer to as the model for simulating a false null hypothesis gene (FNHG) or simply a full model. If the null hypothesis (3) holds, i.e. $\beta_2^C = \beta_{22}^{IC} = \cdots = \beta_{2L}^{IC} = 0$ then the number of parameters corresponding to the fixed effects is reduced to $L$ where the variance-covariance matrix (2) keeps the same. We call this model as that for simulating a null hypothesis gene (NHG) or a reduced model.

Our approach to simulation is to choose one gene with $L$ isoforms, which we refer to as a template gene and fit a full model and a reduced model to the data of the gene. Based on the estimated parameters of the full model and the reduced model we simulate a number of RNA-Seq datasets with $M = 1000$ genes, $L$ isoforms for each gene, and two conditions ($J = 2$).

We conduct the simulation in three scenarios corresponding to three different effect sizes: small, medium and large. In each scenario we simulate data for $m_0$ NHGs and $M-m_0$ FNHGs from multivariate normal distribution based on model (10). To accommodate various patterns of the differentially expressed isoforms in real conditions, we simulate data for two types of FNHGs which we refer to as full FNHGs and partial FNHGs; each accounts for half of the $M-m_0$ genes. The isoforms of a full FNHG are all assumed to be differentially expressed, i.e. the mean difference (9) is not equal to zero for every isoform, whereas a partial FNHG is defined such that only half of the isoforms of gene are differentially expressed. Data for the NHGs are simulated based on the estimated parameters of the reduced model from real data and these parameters are kept the same among the three scenarios with different effect sizes. In generating the data for the FNHGs the parameters $\beta^G$ and $\beta_2^I, \cdots, \beta_L^I$ as well as the parameters for the covariance matrix are also kept the same among the genes and among the three scenarios. The difference lies in the way $\beta_2^C, \beta_{22}^{IC}, \cdots, \beta_{2L}^{IC}$ are specified for the simulations. Suppose that the estimates of these parameters for the full model are $b_2^C, b_{22}^{IC}, \cdots, b_{2L}^{IC}$. If $b_2^C$ is greater than 0, then $\beta_2^C$'s for the full FNHGs are randomly chosen (uniformly) from the intervals $[0, \ b_2^C]$, $[b_2^C - b_2^C/2, \ b_2^C + b_2^C/2]$, and $[b_2^C, \ 2b_2^C]$ corresponding to the small, medium and large effect size scenarios. If $b_2^C$ is less than 0 then the three intervals are $[b_2^C, \ 0]$, $[b_2^C + b_2^C/2, \ b_2^C - b_2^C/2]$, and $[2b_2^C, \ b_2^C]$. $\beta_{22}^{IC}, \cdots, \beta_{2L}^{IC}$ are chosen in the same way. The parameters for simulating partial FNHGs are chosen in the same way except that we force $\beta_{2l}^{IC} = -\beta_2^C$ for $l = [(L+1)/2]+1, \cdots, L$, where $[x]$ is the largest integer not greater than $x$. This will force half of the isoforms not to be differentially expressed. We choose a series of $m_0$ ranging from 0 through $M$, and for each $m_0$ we simulate 100 datasets, each includes $n = 200$ independent sample units for each condition. The level $\alpha$ is set at 0.05 and the simulation results for each $m_0$ are the averages of the results from the 100 replications (simulated datasets).

We applied our two-step procedure to each simulated dataset with Type 1 screening test, two options on the test statistic (t-test and Wald-test) and three options for controlling the FWER (Bonferroni, Holm, and Hochberg methods) that were employed in the confirmatory stage. We compared our proposed two-step procedures to standard isoform-by-isoform analysis with t-tests and BH correction. We called this analysis as simple BH that is borrowed from [32]. There is no gene level screening test for simple B-H method. In order for comparing this method with our two-step procedures we define a screening test for the simple BH method for each gene by rejecting the null hypothesis of no differentially expressed isoforms if the null hypothesis of at least one isoform is rejected by simple BH. We define the OFDR in the same way as shown in (6). To compare the performances of the four procedures we define power (I) and power (II) in the similar way as that in [32]. The power (I) is defined as the proportion of false null hypothesis isoforms that are correctly rejected. It looks at the all $M \times L$ individual isoforms. The power (II) is defined as the proportion of FNHGs that are correctly identified. Here we say that an FNHG is correctly identified if and only if it passes the screening test and a correct decision is made for every single isoform of the gene.

We choose three genes EIF1, MDM2 and ATM as the template genes for the simulations. The numbers of the isoforms for the three genes are 5, 7, 11. The isoform expression profiles of the three genes are presented in supplementary S1 Fig. The unadjusted p-values of the Type 1 screening tests for the three genes are 0.005, 0.123 and 0.003, implying that the data simulated

based on the first and third genes tend to have larger effect sizes than that simulated based the second one if all other simulation settings are the same. Fig 4 presents the evaluation of the OFDR, power(I) and power(II) when Wald-test is used in the confirmatory stage through the simulation with Template Gene MDM2. The results are compared across four methods: our two-step method with Bonferroni, Holm and Hochberg adjustments in the confirmatory stage and the simple BH method. The same simulation results with template genes EIF1 and ATM are shown in the supplementary S2 and S3 Figs. The results for the evaluations when t-test instead of Wald-test is used in the confirmatory stage through the simulation with all three template genes are presented in supplementary S4–S6 Figs.

The three panels in the leftmost column of Fig 4 show that our two-step procedure controls OFDR very well regardless of the effect size and the method used in controlling the FWER in the confirmatory stage; the OFDRs have an increasing trend as the number of NHGs increases with an upper-limit of the nominal level of $\alpha = 0.05$. Fig 4 shows that the OFDR of the simple BH method is substantially inflated and the inflation effect gets serious when the effect size is large. Results of the OFDR are very similar to that of the simulations based on template genes EIF1 and ATM (S2 and S3 Figs.) An over inflated OFDR for simple BH method is seen especially in the simulations based on gene ATM that has the largest number of isoforms among the three template genes (S3 Fig). It implies that there is a positive association between the extent to which the OFDR of simple BH method is inflated and the number of isoforms that each gene possesses.

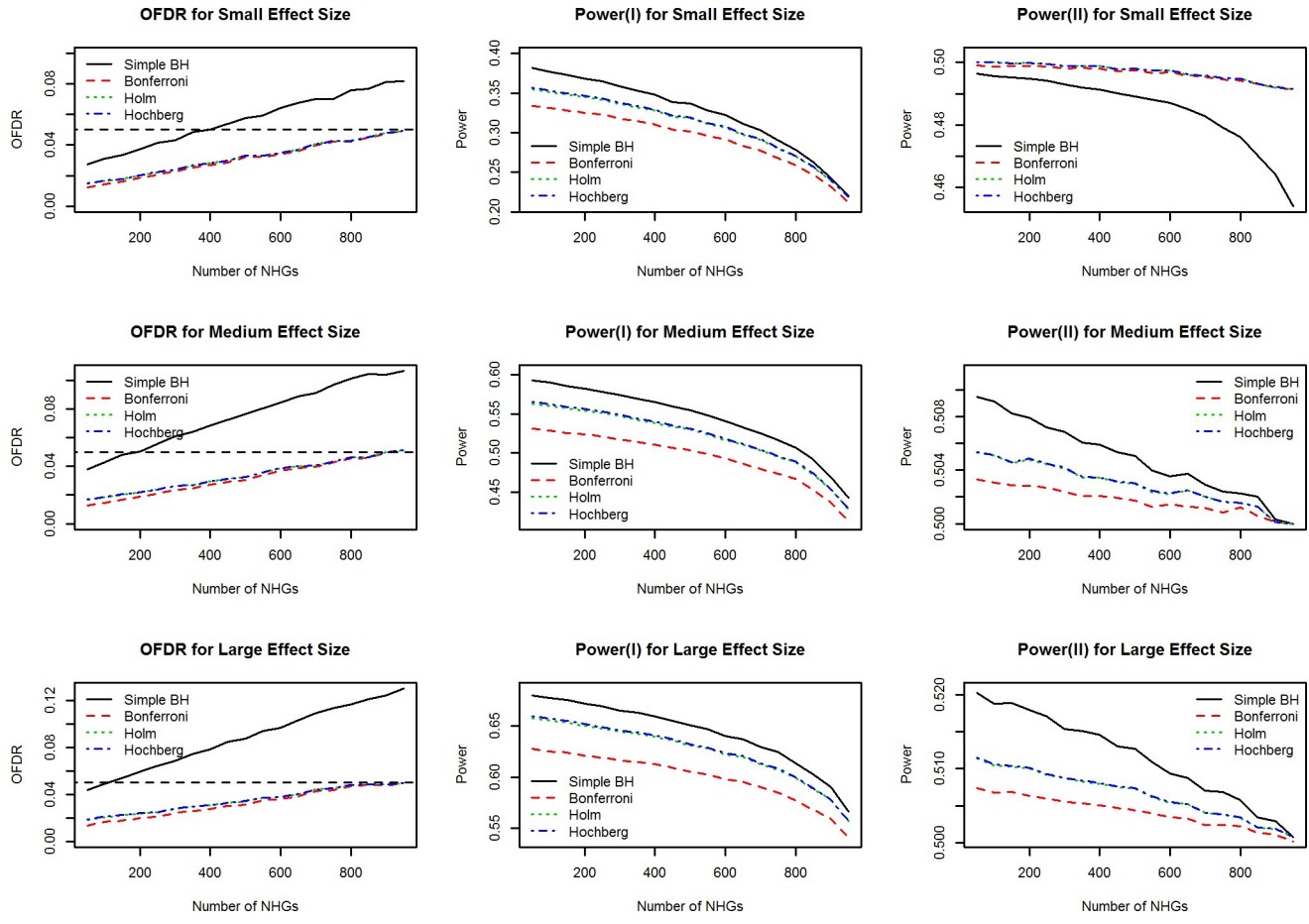

**Fig 4. Evaluation of OFDR and power through the simulation with template gene MDM2 when Wald-test is used in the confirmatory stage.**

The rest six panels in the middle and rightmost of Fig 4 show the simulation results for power (I) and (II), respectively. We can see that our two-step procedure with either Holm or Hochberg adjustment in the confirmatory stage has the same power (I) and (II) regardless of the number of NHGs and the effect sizes, and has improvements in power as compared to that with Bonferroni adjustment. This is what one would expect because the Bonferroni adjustment is more conservative than the other two methods. The results are the same when EIF1 (S2 Fig) or ATM (S3 Fig) is used as the template gene. The simple BH method shows slightly improved power than our two-step procedure in most of settings, which is partly attributable to the over inflated OFDR. When the percentage of NHGs is large (e.g. greater than 90%) the difference in power between the simple BH and our two step procedure is often small. In the setting of small effect size with the simulation based on gene MDM2 (Fig 4) which has the weakest signals among all simulation settings, we observed that our two-step procedure shows an improved power (II) compared to the simple BH method although the OFDR of simple BH is still overly inflated. The difference in power (II) substantially increases as the number of NHGs increases. It implies that when the effect sizes are small and the number of NHGs are large (which is common in real data sets) our method provides more statistical power than the simple BH. This partially explains why in the real data example the simple BH method could not identify any significant isoforms.

We examined the differences caused by using different tests (Wald-test or t-test) in the confirmatory stage by comparing Fig 3 and S2 and S3 with S4–S6 Figs. The OFDR is well controlled regardless of whether the Wald-test or the t-test is used in the confirmatory stage and the differences in powers are negligible for all the simulation settings.

We repeat the evaluation of OFDR and power of our two-step procedure and simple BH under a setting of a small sample size, $n = 50$ for each condition (S7–S9 Figs). As we expected the power of all the methods is reduced. The power of the simple BH method has more reduction as compared to our two-step procedure while the OFDR of the simple BH keeps inflated in all the settings. We also notice that when the percentage of NHGs is large, the OFDR of our two-step procedure is also slightly inflated. This is because the likelihood ratio tests using the standard Chi-square distribution for fixed effects in linear mixed effects models tend to be "anticonservative" and their type I errors are sometimes inflated, which has been reported in the literature [46]. To overcome this limitation one may use the F-test with Kenward-Roger approximation or parametric bootstrap methods [46] instead of using likelihood ratio test in the screening test.

## Discussion

High-throughput RNA-Seq technologies have provided the unprecedented opportunity to characterize whole-transcriptome profiling of gene expression, and in addition facilitates quantification of complex alternative splicing patterns. Systematically investigating the relationship between the complex alternative splicing patterns and the biological or clinical patient outcomes requires advanced statistical method and tools that appropriately account for the unique data characteristics, e.g. differential expression/splicing patterns. In this work, we developed a unified approach for simultaneously assessing genes with differentially expressed or spliced isoforms, which also employed appropriate hierarchical hypotheses-testing framework to efficiently alleviate the multiple comparison burden.

The multi-dimensional structure of gene- and isoform- level expression induces extremely high burden of multiple comparisons, where appropriate corrections for multiple comparisons need special care. Traditional statistical tools focusing on corrections for each individual isoform or all possible pairwise comparisons of two isoforms will yield limited statistical power.

The stage-wise method (stageR) developed by Van den Berge et al. [27] utilized the two-step hierarchical hypothesis testing framework to determine the gene-level and isoform-level significance in a two-step fashion, which represent great efforts to overcome such limitations. However, the limitation in the stageR method lies in the statistical methods used to determine the gene-level significance in the first step. They rely on existing statistical methods of evaluating the aggregated transcript level differential expression or different usage/switches of transcript expression to determine the gene-level significance. These approaches are often based on generalized linear models, do not distinguish the genes with different patterns of expression and splicing, and do not account for the correlation between isoforms from the same gene as well as the random subject effects. To build upon existing methods and address the analytical challenges particularly in effectively detecting significant genes that are differentially expressed or spliced, we developed the linear mixed effects model framework for analysis of the complex alternative RNA splicing patterns.

Our approach provides several advantages in modeling the patterns of differential expression/splicing while appropriately accounts for the correlation among different isoforms. First, the utilized linear model approach fully characterizes three different types of genes with distinct alternative RNA splicing patterns with differential expression and or differential splicing profiles. We proposed two types of screening tests using a linear mixed effects model approach that thoroughly characterizes and differentiates three types of genes related to alternative RNA splicing events. Specifically, our approach differentiates among genes that have no differentially expressed isoforms, genes that have differential expression of isoforms but no differential splicing, and genes with differentially spliced isoforms with differential expression at the isoform level but not necessarily at the gene level. Our proposed type 1 screening tests will identify genes that have either differential isoform expression or differential isoform switches at the same time. The significant genes detected using the type 1 screening tests provide a solid foundation for downstream pathway analysis in order to identify biologically meaningful gene sets that are enriched for genes with different types of alternative splicing events. This overcomes the limitation of existing pathway-based alternative splicing analysis which requires an extra step to combine the significance in differential expression and splicing [47]. Second, our mixed model including fixed and random effects effectively accounts for the biological correlation structure among isoforms of the same gene. These correlations are often times ignored in other existing statistical tools, which may yield false positive or false negative results. Third, we conducted extensive simulation studies and demonstrated that our proposed linear mixed effects modeling framework coupled with the use of two-step hierarchical hypothesis testing procedure appropriately controls the gene-level overall false discovery rate (OFDR) which also provides improved statistical power and computational efficiency to discover genes with significant differential expression/splicing patterns or genes with isoform switches. Lastly, the application to two real RNA-Seq studies have demonstrated the advantages and improved performances of our method in the differential alternative RNA splicing analysis.

As with other studies where a linear mixed effects model is applied, a typical issue is how to choose the covariance structure. We have compared and provided the analysis scripts for three different choices of covariance matrix (compound symmetry with equal variances, unequal variances, and unstructured). We focus on demonstrating the results from unequal variances covariance structures in the main text, but also included the results from other covariance structures for open access through http://www.unm.edu/~kanghn/software/. We note the limitations of the two other choices. The compound symmetry with equal variances covariance structure is limited in capturing the different variances in the different isoform expressions. The unstructured covariance structure covariance has no constraints imposed on it which results in best model fit, however, it will suffer from inflated type I error for relatively small

samples as previously reported [48]. In the case of large sample sizes, the computation time for incorporating an unstructured covariance structure is much larger than other choices of covariance structures. The unequal variances covariance structure is in the middle of these two covariance structures. It can be used more broadly because not only does it allow for heterogeneous variances among the isoforms, it also captures the random interaction effects between the subjects and isoforms. It is also a parsimonious model as it has a fewer number of parameters than the unstructured covariance does. We observed substantial overlap of genes on the top of the list ranked based on the p-values of the first-step screening tests obtained using unequal variance and unstructured covariance matrix. For those non-overlapping genes, i.e. genes passed the screening test in the first step using the unequal variance covariance structures but not using the unstructured covariance matrix, we observed that that false discovery rates for these genes using the unstructured covariance matrix are also small (median FDR of 0.067–0.129). The correlations between the p-values for gene-based screening tests are high with Spearman's $\rho$ of 0.92–0.94. We focused on the discussion of our results using the covariance structure with unequal variances from the practical perspective, which accounts for both the computation time and the appropriate covariance structure assumption that suits the majority of genes. However, we also provided the alternative option of using unstructured covariance matrix, which can be adapted for specific case scenarios in other applications.

In summary, we developed a unified approach for simultaneously assessing genes with differentially expressed or spliced isoforms that offers much more flexible and improved performance than existing methods which can 1) incorporate a broader range of models to analyze differential splicing and expression; 2) include testing between conditions that have multiple samples within each condition and allow more than two conditions; 3) maintain statistical power with proper control of OFDR; and 4) appropriately control for confounding clinical factors which are common in large genetic epidemiological studies.

## Supporting information

**S1 Fig. The isoform expression profiles for three template genes used for simulation study.**
(PDF)

**S2 Fig. Evaluation of OFDR and power through the simulation with template gene EIF1 when Wald-test is used in the confirmatory stage.**
(PDF)

**S3 Fig. Evaluation of OFDR and power through the simulation with template gene ATM when Wald-test is used in the confirmatory stage.**
(PDF)

**S4 Fig. Evaluation of OFDR and power through the simulation with template gene EIF1 when t-test is used in the confirmatory stage.**
(PDF)

**S5 Fig. Evaluation of OFDR and power through the Simulation with Template Gene MDM2 when t-test is used in the confirmatory stage.**
(PDF)

**S6 Fig. Evaluation of OFDR and power through the simulation with template gene ATM when t-test is used in the confirmatory stage.**
(PDF)

**S7 Fig. Simulation with template gene EIF1 and small sample size (N = 50 in each group) when Wald-test is used in the confirmatory stage.**
(PDF)

**S8 Fig. Simulation with template gene MDM2 and small sample size (N = 50 in each group) when Wald-test is used in the confirmatory stage.**
(PDF)

**S9 Fig. Simulation with template gene ATM and small sample size (N = 50 in each group) when Wald-test is used in the confirmatory stage.**
(PDF)

**S1 Table. List of genes that were passed type 1 and 2 screening tests along with the likelihood ratio test p-values and FDR.** Type 1 screening test identified 50 genes with differentially expressed/spliced isoforms, and type 2 screening test identified 8 genes with differentially spliced genes. Results were obtained from linear mixed model with unstructured covariance structure for ACC Study.
(PDF)

**S2 Table. List of 18 differentially expressed isoforms between 8 patients who are free of cancer vs. 6 patients in the confirmatory test.** Results obtained from the Wald test following the type 1 screening test based on the linear mixed model with unstructured covariance structure for ACC Study.
(PDF)

**S3 Table. List of genes that were passed type 1 and 2 screening tests along with the likelihood ratio test p-values and FDR.** Type 1 screening test identified 782 genes with differentially expressed/spliced isoforms, and type 2 screening test identified 857 genes woth differentially spliced genes. Results were obtained from linear mixed model with unequal variance covariance structure for AML Study.
(PDF)

**S4 Table. List of differentially expressed isoforms based on type 1 screening test for AML study.**
(PDF)

**S5 Table. List of differentially expressed isoforms based on type 2 screening test for AML study.**
(PDF)

## Author Contributions

**Conceptualization:** Li Luo, Huining Kang, Scott A. Ness, Christine A. Stidley.

**Data curation:** Li Luo, Huining Kang, Xichen Li.

**Formal analysis:** Li Luo, Huining Kang, Xichen Li.

**Funding acquisition:** Scott A. Ness.

**Investigation:** Huining Kang, Scott A. Ness.

**Methodology:** Li Luo, Huining Kang, Christine A. Stidley.

**Software:** Li Luo, Huining Kang.

**Visualization:** Li Luo, Huining Kang, Christine A. Stidley.

**Writing – original draft:** Li Luo, Huining Kang.

**Writing – review & editing:** Li Luo, Huining Kang, Xichen Li, Scott A. Ness, Christine A. Stidley.

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
