## [Decision Letter · Decision Letter 0]

18 May 2020

PONE-D-20-11155

Two-Step Mixed Model Approach to Analyzing Differential Alternative RNA Splicing

PLOS ONE

Dear Dr. Kang,

Thank you for submitting your manuscript to PLOS ONE. After careful consideration, we feel that it has merit but does not fully meet PLOS ONE’s publication criteria as it currently stands. Therefore, we invite you to submit a revised version of the manuscript that addresses the points raised during the review process.

We would appreciate receiving your revised manuscript by Jul 02 2020 11:59PM. To enhance the reproducibility of your results, we recommend that if applicable you deposit your laboratory protocols in protocols.io, where a protocol can be assigned its own identifier (DOI) such that it can be cited independently in the future. For instructions see: http://journals.plos.org/plosone/s/submission-guidelines#loc-laboratory-protocols

We look forward to receiving your revised manuscript.

Kind regards,

Zhaohui Qin, PhD

Academic Editor

PLOS ONE

Journal Requirements:

Reviewers' comments:

Reviewer's Responses to Questions

**Comments to the Author**

1. Is the manuscript technically sound, and do the data support the conclusions?

Reviewer #1: Yes

Reviewer #2: Yes

2. Has the statistical analysis been performed appropriately and rigorously? 

Reviewer #1: Yes

Reviewer #2: Yes

3. Have the authors made all data underlying the findings in their manuscript fully available?

Reviewer #1: Yes

Reviewer #2: Yes

4. Is the manuscript presented in an intelligible fashion and written in standard English?

Reviewer #1: Yes

Reviewer #2: Yes

5. Review Comments to the Author

Reviewer #1: The authors applied a two-stage hypothesis testing procedure for testing differentially expressed or spliced isoforms, within a linear mixed model setting. The novelty of the methodology part is limited, as this work can be regarded as an extended application to RNA-seq data, compared to Heller (2009) and Li (2014) where the two-step testing procedure was originally proposed. Nevertheless, the data experiments of this paper were solid and complete, allowing this paper to be a good application report with interesting scientific findings.

Minor comment:

It is appreciative that the authors provided code for this work. A gentle advice would be providing a demonstration of applying the R functions (i.e. pipeline), rather than the functions only.

Reviewer #2: The authors propose to use a linear mixed model to simultaneously assess genes with differentially expressed or spliced isoforms. A two-step hypothesis testing procedure is proposed to select the significant genes and isoforms, which controls the gene-level overall false discovery rate. The proposed method is novel and enjoys some advantages over the existing methods. The authors further demonstrate the performance of the proposed method using two real data sets and a thorough simulation study.

I only have one comment: the validation of the proposed method relies on the validity of the linear mixed model (1), where the covariance matrix of y has equal off-diagonal elements in (2). Some comments are needed for the validation of such a model. If the assumption of the covariance matrix is violated in applications, the model (1) is not applicable. Is the proposed method robust to use in such a scenario? What modification is need?

6. PLOS authors have the option to publish the peer review history of their article (what does this mean?). If published, this will include your full peer review and any attached files.

Reviewer #1: No

Reviewer #2: No

---

## [Author Response · Author response to Decision Letter 0]

6 Sep 2020

PONE-D-20-11155: “Two-Step Mixed Model Approach to Analyzing Differential Alternative RNA Splicing”

Dear Dr. Qin,

Thank you for reviewing our manuscript. We have made revisions in response to the reviewers’ comments. A marked-up copy of the manuscript highlighting the changes as well as a clean copy of the manuscript without tracked changes are also uploaded. We also made additional edits to ensure that the manuscript meets PLOS ONE’s style requirements. In this response letter, we have included each of the referees’ comments followed by our reply (Response).

Reviewer 1: 

Comment: The authors applied a two-stage hypothesis testing procedure for testing differentially expressed or spliced isoforms, within a linear mixed model setting. The novelty of the methodology part is limited, as this work can be regarded as an extended application to RNA-seq data, compared to Heller (2009) and Li (2014) where the two-step testing procedure was originally proposed. Nevertheless, the data experiments of this paper were solid and complete, allowing this paper to be a good application report with interesting scientific findings.

Response: We appreciate the reviewer’s comment and have added a brief discussion/clarification of this study. Please see the edits on page 5 in the revised manuscript with track changes document.

“We present a novel application of two-step hierarchical hypothesis testing procedure coupled with a linear mixed model setting to analyze the differential expression at the multi-dimensional gene- and isoform- level using data from RNA sequencing technologies. ”

Minor comment: It is appreciative that the authors provided code for this work. A gentle advice would be providing a demonstration of applying the R functions (i.e. pipeline), rather than the functions only.

Response: We have prepared a tutorial in the html format generated through the use of R markdown file illustrating the analytical pipeline. The tutorial that describe the step-by-step instructions on using the collection of R functions is available through http://www.unm.edu/~kanghn/software/Tutorial.html and the Dryad Digital Repository (doi:10.5061/dryad.66t1g1k0h). We have also provided two documents in html format to explain the detailed steps and R codes for analyses of the two example data. Please see edits in the section of Availability of data and materials on page 27 in the revised manuscript with track changes document. 

Reviewer 2: 

The authors propose to use a linear mixed model to simultaneously assess genes with differentially expressed or spliced isoforms. A two-step hypothesis testing procedure is proposed to select the significant genes and isoforms, which controls the gene-level overall false discovery rate. The proposed method is novel and enjoys some advantages over the existing methods. The authors further demonstrate the performance of the proposed method using two real data sets and a thorough simulation study.

Comment: I only have one comment: the validation of the proposed method relies on the validity of the linear mixed model (1), where the covariance matrix of y has equal off-diagonal elements in (2). Some comments are needed for the validation of such a model. If the assumption of the covariance matrix is violated in applications, the model (1) is not applicable. Is the proposed method robust to use in such a scenario? What modification is need?

Response: We appreciate the reviewer’s comment and agree that this is a good point. We provided analyses scripts that incorporating three different choices of covariance matrix (compound symmetry with equal variances, unequal variances, and unstructured), and compared the results in the applications and added discussions on the choices of covariance matrix. If the assumption that the covariance matrix of y has equal off-diagonal is violated, we suggest using the unstructured covariance structure that has no constraints on the parameters resulting in the best model fit. However, we also note some limitations of using the unstructured covariance structure. It will suffer from inflated type I error for relatively small samples as previously reported by Kwok et al. 2007. In the case of large sample sizes, the computation time for incorporating an unstructured covariance structure is much larger than other choices of covariance structures. 

The compound symmetry with equal variances covariance structure, on the other hand, is limited in capturing the different variances in the different isoform expressions. The unequal variances covariance structure resides in the middle of these two covariance structures. It can be used more broadly because not only does it allow for heterogeneous variances among the isoforms, it also captures the random interaction effects between the subjects and isoforms. It is also a parsimonious model as it has a fewer number of parameters than the unstructured covariance does. We focused on the discussion of our results using the covariance structure with unequal variances from the practical perspective, which accounts for both the computation time and the appropriate covariance structure assumption that suits the majority of genes. 

Please see edits in the results section (page 12, 17) and discussion section (pages 25-26) in the revised manuscript with track changes document. 

Kwok O-m, West SG, Green SB. The Impact of Misspecifying the Within-Subject Covariance Structure in Multiwave Longitudinal Multilevel Models: A Monte Carlo Study. MULTIVARIATE BEHAVIORAL RESEARCH. 2007;42(3):557-92.

Additional Comments on Journal Requirements:

Comment: 1) We note that your manuscript is not formatted using one of PLOS ONE’s accepted file types. Please reattach your manuscript as one of the following file types: .doc, .docx, .rtf, or .tex (accompanied by a .pdf). If your submission was prepared in LaTex, please submit your manuscript file in PDF format and attach your .tex file as “other.”

Response: We have reattached the manuscript files as the .docx file type. 

Comment: 2) Unfortunately, the repository you have noted in your Data Availability statement does not qualify as an acceptable data repository according to PLOS ONE's standards, as it is public but does not appear to be stable. At this time, please upload the minimal data set necessary to replicate your study's findings to a stable, public repository (such as figshare or Dryad) and provide us with the relevant URLs, DOIs, or accession numbers that may be used to access these data.

Response: We have uploaded the datasets and analysis scripts for replicating our study findings to the Dryad Digital Repository (doi:10.5061/dryad.66t1g1k0h). The datasets are currently for private access during this review period, which can be accessed through: https://datadryad.org/stash/share/yRDf1Kmj9_hR_I-IGg_vukBVNUmmB9tm_j8v1BZ721A. We have also revised the Data Availability statement to include the Dryad doi. 

Sincerely,

Li Luo, Ph.D. and Huining Kang, Ph.D.

University of New Mexico Comprehensive Cancer Center

Division of Epidemiology, Biostatistics, and Preventive Medicine

Department of Internal Medicine, University of New Mexico

Emails: LLuo@salud.unm.edu and HuKang@salud.unm.edu

---

## [Decision Letter · Decision Letter 1]

25 Sep 2020

Two-Step Mixed Model Approach to Analyzing Differential Alternative RNA Splicing

PONE-D-20-11155R1

Dear Dr. Kang,

We’re pleased to inform you that your manuscript has been judged scientifically suitable for publication and will be formally accepted for publication once it meets all outstanding technical requirements.

Kind regards,

Zhaohui Qin, PhD

Academic Editor

PLOS ONE

Additional Editor Comments (optional):

Reviewers' comments:

Reviewer's Responses to Questions

**Comments to the Author**

1. If the authors have adequately addressed your comments raised in a previous round of review and you feel that this manuscript is now acceptable for publication, you may indicate that here to bypass the “Comments to the Author” section, enter your conflict of interest statement in the “Confidential to Editor” section, and submit your "Accept" recommendation.

Reviewer #1: All comments have been addressed

Reviewer #2: All comments have been addressed

2. Is the manuscript technically sound, and do the data support the conclusions?

Reviewer #1: Yes

Reviewer #2: Yes

3. Has the statistical analysis been performed appropriately and rigorously? 

Reviewer #1: Yes

Reviewer #2: Yes

4. Have the authors made all data underlying the findings in their manuscript fully available?

Reviewer #1: Yes

Reviewer #2: Yes

5. Is the manuscript presented in an intelligible fashion and written in standard English?

Reviewer #1: Yes

Reviewer #2: Yes

6. Review Comments to the Author

Reviewer #1: The authors have addressed my comments in the first round of review, particularly, they prepared a much more user-friendly tutorial regarding code usage. I don't have further comments for this submission.

Reviewer #2: (No Response)

7. PLOS authors have the option to publish the peer review history of their article (what does this mean?). If published, this will include your full peer review and any attached files.

Reviewer #1: No

Reviewer #2: No

---

## [Editor Report · Acceptance letter]

29 Sep 2020

PONE-D-20-11155R1 

Two-Step Mixed Model Approach to Analyzing Differential Alternative RNA Splicing 

Dear Dr. Kang:

I'm pleased to inform you that your manuscript has been deemed suitable for publication in PLOS ONE. Congratulations! Your manuscript is now with our production department. 

Kind regards, 

on behalf of

Associate Professor Zhaohui Qin 

Academic Editor

PLOS ONE